# Dichotomy of cellular inhibition by small-molecule inhibitors revealed by single-cell analysis

Robert M. Vogel[1], Amir Erez[1,†] & Grégoire Altan-Bonnet[1,†]

Despite progress in drug development, a quantitative and physiological understanding of how small-molecule inhibitors act on cells is lacking. Here, we measure the signalling and proliferative response of individual primary T-lymphocytes to a combination of antigen, cytokine and drug. We uncover two distinct modes of signalling inhibition: digital inhibition (the activated fraction of cells diminishes upon drug treatment, but active cells appear unperturbed), versus analogue inhibition (the activated fraction is unperturbed whereas activation response is diminished). We introduce a computational model of the signalling cascade that accounts for such inhibition dichotomy, and test the model predictions for the phenotypic variability of cellular responses. Finally, we demonstrate that the digital/analogue dichotomy of cellular response as revealed on short (signal transduction) timescales, translates into similar dichotomy on longer (proliferation) timescales. Our single-cell analysis of drug action illustrates the strength of quantitative approaches to translate *in vitro* pharmacology into functionally relevant cellular settings.

[1] ImmunoDynamics Group, Program in Computational Biology and Immunology, Memorial Sloan Kettering Cancer Center, 1275 York Avenue, Box 460, New York, New York 10065, USA. † Present address: ImmunoDynamics Group, Cancer & Inflammation Program, Center for Cancer Research—National Cancer Institute, Bldg 37—Room 4134B, 37 Convent Drive, Bethesda, Maryland 20892, USA (A.E. and G.A.-B.). Correspondence and requests for materials should be addressed to G.A.-B. (email: gregoire.altan–bonnet@nih.gov).

ndividual cells rely on biochemical signalling pathways to translate environmental cues into physiological responses. Spurious activation of these pathways results in a cell's mischaracterization of environmental conditions and aberrant cellular behaviour. This behaviour can, in some cases, be detrimental to the health of the organism—causing ailments such as inflammatory diseases (for example, ulcerative colitis[1]), auto-immune disorders[2,3] and cancer[4]. Inhibiting specific dysfunctional components with small-molecule chemical inhibitors has been successful in reducing aberrant signals and ultimately ailments[5]. Examples include Imatinib in treating chronic myelogenous leukaemia[6] and Gefitnib for patients with EGFR mutant non-small-cell lung cancer[7,8]. However, despite these successes, inhibitory drug development remains slow and can benefit from new techniques to aid screening of candidate compounds[9,10].

Fundamentally, an effective chemical inhibitor acts on a signalling pathway by binding to the targeted enzyme and shutting down its enzymatic activity. In this context, optimizing a drug inhibitor abounds to optimizing its specific binding to the enzyme target of choice. Recent technological advances have focused efforts to development of pipelines that characterize drug specificity with respect to all human kinases *in vitro*[11–13] and in cell lines as models for physiological settings[14]. The emphasis on protein kinases is due to their prominence in signal transduction pathways, where they serve as information relays by transferring a phosphate from ATP to their target substrate. The technological advances in drug screening often fall short of anticipating the downstream consequences of drug inhibition: whereas kinase inhibition is optimized at the local (molecular) level, the response at the level of the entire pathway often remains sub-optimal. Consequently, it is difficult to predict cellular response to chemical inhibition. To gain some understanding of this response, emphasis has been placed on high-throughput characterization of the response of cell lines[15,16].

Despite these advances in drug screening, many poorly performing compounds proceed to, and often fail at, the organismal stage of drug discovery. This suggests that we may need to re-evaluate the relevance of bulk measurements on cell line models to drug development, emphasizing instead a more mechanistic understanding of individual primary cell responses to inhibition ('What's wrong with drug screening today'[17]). This need has been partially addressed by pioneering studies that characterized biochemical networks of primary cells[18,19] and canonical cell-type responses to inhibition[20–22]. Yet, while these studies have been illuminating, mechanistic principles of cellular responses to small-molecule chemical inhibition have remained elusive. It is precisely this gap in knowledge that this communication attempts to address.

We conjecture that one needs to resolve diverse enzymatic states (for example, phospho-status) at the single-cell level to identify the complex nonlinear responses of signalling networks to inhibition. Nonlinear responses are often dominated by a subset of enzymes that determine the behaviour of the pathway. Identifying these key enzymes uncovers novel vulnerabilities of the signalling network to inhibition[23]. Examples of nonlinear responses uncovered by single-cell measurements are numerous: flow cytometry measurements of double-phosphorylated ERK (ppERK) accumulation in stimulated T-lymphocytes exhibit a highly nonlinear bimodal response to antigen[24]; by imaging ERK in live cells, individual cell response to growth factors was shown to be pulsatile[25] or oscillatory[26]; administration of either the tyrosine kinase inhibitor Gefitnib or the MEK inhibitor PD325901 yielded either a frequency or a mean reduction in ppERK signalling, respectively[27]. These are but few examples of the dynamic complexities of biochemical signalling networks

under stimulation, as revealed by single-cell measurements. In all these examples, inaccessible by 'high-throughput' population level measurements, single-cell measurements added crucial understanding to the structure of a signalling pathway.

In this study, we integrate single-cell multi-parametric phospho-flow cytometry measurements, cell-to-cell variability analysis (CCVA[28,29]), and mechanistic modelling into a single framework for dissecting the response of primary T-lymphocytes to kinase inhibitors. By applying our integrated framework, we find that protein variability creates diverse sensitivities of individual cells to inhibition in two unique signalling systems, namely JAK–STAT and T-cell receptor (TCR)-mediated mitogen-activated protein kinase (MAPK) pathway. These observations provide the necessary, and rarely utilized, constraints required for mechanistic insights of kinase inhibition in a physiological setting. We formally incorporate these insights into mechanistic mathematical models, of which are available in Supplementary Notes 1 and 2. We find that (i) JAK inhibitor AZD1480 functions as a noncompetitive inhibitor with STAT5, (ii) SRC (sarcoma kinase) and MEK (MAPK/ERK kinase) inhibitors exhibit two qualitatively different modes of inhibition. Specifically, targeting SRC manifests into either maximal or unmeasurable quantities of phosphorylated ERK (digital); conversely, inhibiting MEK produces a graded response (analogue). With these constraints we developed a coarse-grained model of TCR signalling containing exclusively measurable quantities and the targets of the respective inhibitors. Using this model we determined that the disparate response of cells to these inhibitors originates in the unique embeddings of SRC and MEK in biological networks. Following these insights we demonstrate the functional relevance of our model of inhibition to cell proliferation, thereby bridging the short molecular timescale with the longer functional one. Taken together, in this communication we demonstrate how, by combining mechanistic models and single-cell measurements of primary cells, it is possible to predict markedly different cellular behaviour in response to targeted molecular inhibition.

## Results

**Single-cell diversity originates from protein variability**. A reductionist approach posits that the properties of a signal transduction pathway in living cells should be deductible from the biochemistry of its components working in concert. However, traditional methods such as *in vitro* assays of enzyme extracts and ensemble average measurements (for example, western blot) do not incorporate the inherent biological complexity of cells or the required resolution, and therefore fall short of a detailed biochemical characterization of chemical inhibitors. To illustrate this issue, we investigated the biochemistry of JAK-induced STAT5 phosphorylation in individual T lymphocytes stimulated with the cytokine interleukin 2 (IL-2, Fig. 1a). We focus on this pathway for three reasons: (i) its biological function is important, corresponding to anti-apoptotic and proliferative signals[30]; (ii) its clinical relevance in inflammatory diseases[2,3] and cancer[31]; (iii) its the molecular components are well documented[32].

To monitor the JAK/STAT signalling response to JAK inhibition—we prepared *ex vivo* mouse primary T-cell blasts and exposed them to saturating amounts of the cytokine IL-2 (2 nM), followed by two-fold serial dilutions of AZD1480 ($I_{JAK}$), and measured at steady state (Supplementary Note 1.1). We found that the average response follows an inhibitory hill function with an estimated half effective inhibition concentration ($IC_{50}$) of $8.2 \pm 0.5$ nM (Fig. 1b). In this preliminary characterization we assumed that the hill coefficient is exactly one. A hill

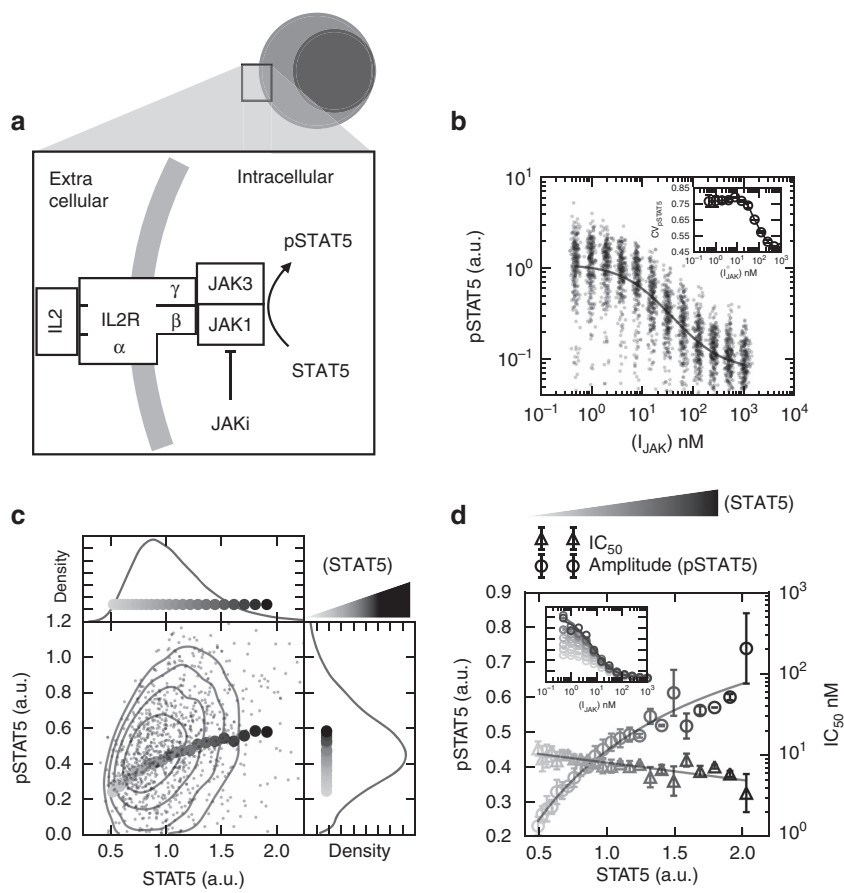

**Figure 1 | Variability of endogenous protein abundance correlates with single-cell response to chemical inhibition.** (**a**) IL-2 stimulation of the JAK–STAT pathway. (**b**) Single-cell pSTAT5 abundance in response to JAK inhibitor AZD1480. Inset, the coefficient of variation (CV) response to inhibition. (**c**) Single-cell contour plot of total STAT5 abundance and pSTAT5 in cells not treated with inhibitor, $[I_{JAK}] = 0$. Curve shows the resulting geometric mean of the pSTAT5 abundance conditioned on STAT5 abundance per cell. (**d**) Cell-to-Cell variability analysis reveals that the pSTAT5 response amplitude is correlated with STAT5 abundance. In addition, the sensitivity of cells to inhibition ($IC_{50}$) exhibits a small negative correlation with STAT5 abundance (errorbars are standard deviation of experimental duplicates).

coefficient of one indicates that the inhibition of STAT5 phosphorylation can be described by the drug simultaneously binding and deactivating the kinase.

The phospho-STAT5 (pSTAT5) response of individual cells to JAK inhibition decreases smoothly and unimodally with increasing doses of drug (Fig. 1b). We characterized the variability of cell responses by the coefficient of variation (CV), a measure of the standard deviation with respect to the mean pSTAT5 response. In the absence of drug the CV is $0.77 \pm 0.004$, and depreciates with increasing doses of inhibitor (Fig. 1b inset). The concomitant decrease in the mean response and $CV_{pSTAT5}$ contradicts the stochastic properties of chemical reactions. Indeed, diversity in the abundance of pSTAT5 originating from physico-chemical mechanisms is expected to exhibit Poisson statistics, meaning that the $CV_{pSTAT5}$ should behave as the inverse square root of the mean[33,34]. Therefore, in contrast to our observations, if the origin of the noise were Poissonian, $CV_{pSTAT5}$ originating from these simple Poisson properties would increase, rather than decrease, with increasing inhibitor dosage. Consequently, we conclude that individual clones generate diverse levels of pSTAT5 from biological sources of diversity, that is, protein variability, as opposed to the intrinsic stochasticity of chemical reactions.

Next we asked whether the variable abundance of STAT5 can explain pSTAT5 variability in response to JAK inhibition. We simultaneously monitored both STAT5 and pSTAT5 in individual cells, and measured the average pSTAT5 abundance in subpopulations of cells with similar STAT5 abundances, a technique referred to as CCVA[28,29]. We found that the geometric mean of pSTAT5 correlates with STAT5 abundance in the absence of JAK inhibitor (Fig. 1c). We then investigated how varying abundances of STAT5 influence both the JAK inhibitor dose response amplitude and the half effective inhibitor concentration ($IC_{50}$). We found that the amplitude of pSTAT5 response increased with STAT5 expression, while the $IC_{50}$ reduced exponentially with a scale of approximately $-2.0$ (STAT5 a.u., Fig. 1d). Hence, by monitoring the extent of drug inhibition at the single-cell level, we establish new experimental observations regarding signal inhibition.

CCVA established the dependence of pSTAT5 on the endogenous (variable) STAT5 abundance. We leveraged this observation to develop a biochemical model of inhibition in live cells. Specifically, we tested three simple biochemical models that may account for the transmission of STAT5 variability to pSTAT5 levels per cell, and the biochemical mechanism of JAK inhibition by AZD1480 (Fig. 2a, see Supplementary Notes 1.2–1.4 for derivations). Each mechanism represents unique interactions between the JAK, STAT5 and IJAK—the noncompetitive inhibitor binds to JAK independent to the presence of STAT5, the uncompetitive inhibitor only binds to the JAK–STAT5 complex, and the competitive inhibitor binds to JAK which prevents STAT5 from binding (Fig. 2a). We found

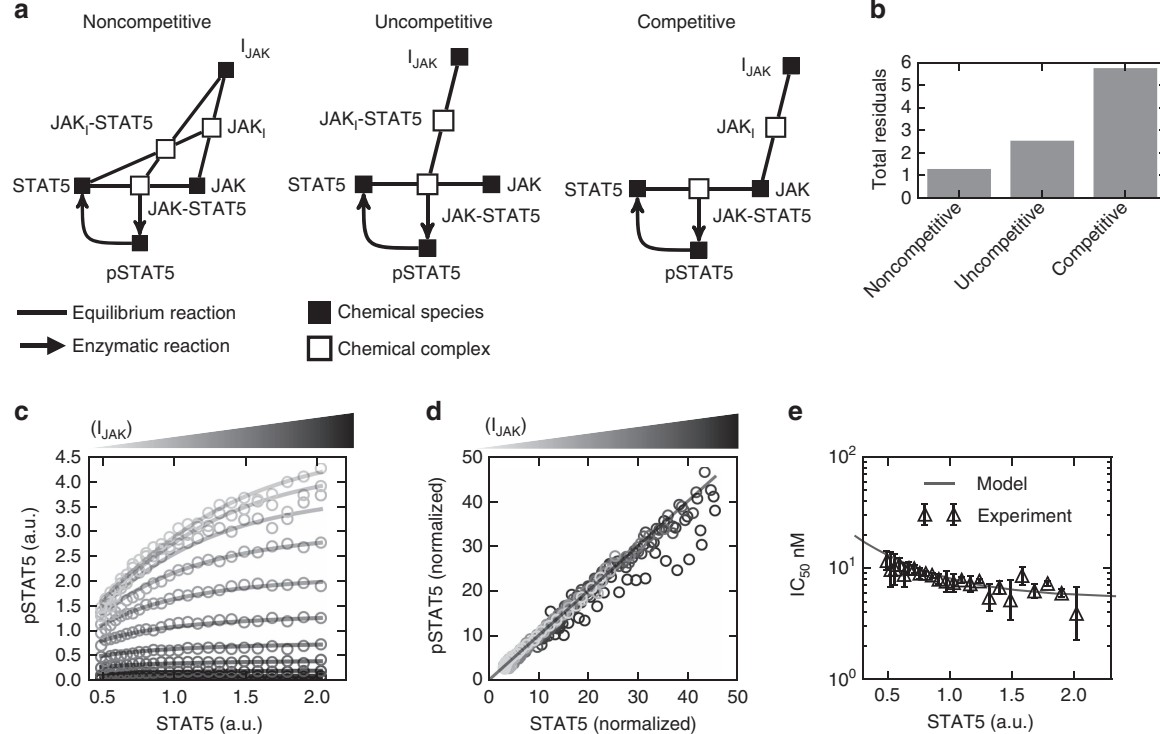

**Figure 2 | CCVA reveals the most likely mechanism of AZD1480 in live single cells.** (**a**) Model diagrams that represent three possible mechanisms of inhibition. (**b**) Each model was tested against our data by measuring the sum of squared residuals (total residuals) between our model predictions and the data points—a lower value means better agreement between model and data. The model was fit to all the data point presented in **c**. (**c**) Overlay of data (circles) with the optimal model and parameter set from fit (lines). (**d**) The linearized data (Supplementary Equation 8) was derived from the optimal model and the corresponding parameters reveal agreement between model (line) and the data (open circles). (**e**) Overlay of measured $IC_{50}$ with respect to STAT5 abundance as measured from CCVA analysis of data (triangles; errorbars standard deviation experimental duplicates) and predicted by our optimal model (line).

that a noncompetitive inhibition model for AZD1480 action best described our experimental observations (Fig. 2b, see Supplementary Note 1.5 for details). This observation is supported by the fact that AZD1480 acts by competing with ATP for occupancy of the ATP binding pocket of JAK, and does not compete with STAT5 (ref. 35). Furthermore, it was necessary to account for the physiological variability in STAT5 substrate availability to account for the cell-to-cell variability in pSTAT5 inhibition (Fig. 2c,d). Lastly, we validated that our model could account for the small dependence of the $IC_{50}$ on STAT5 expression. We found agreement between our $IC_{50}$ measurements in Fig. 1d with the estimated $IC_{50}$ from our model (Fig. 2e).

To summarize, in this section we demonstrated how CCVA parses single-cell phospho-profiling data to validate models of drug inhibition. We employed CCVA here on the JAK–STAT pathway, and found an optimal model of noncompetitive binding of inhibitor to kinase, as supported by the literature.

**Single-cell measurements reveal diverse modes of inhibition.** We proceed by investigating inhibition of a more complex signalling cascade, namely antigen-driven MAP kinase activation in primary T cells. Upon exposure with activating ligands (for example, complex of a peptide with major histocompatibility complex, peptide-MHC presented on the surface of antigen-presenting cells (APCs)), T cells activate their receptors through activation of a SRC Family kinase (Lck). This then triggers a cascade of kinase activation leading to ERK phosphorylation. We chose this model system because its complex network topology and its functional relevance: aberrant activation in the ERK pathway is often involved with oncogenesis[4], making it a key

pathway to be targeted with drug inhibitors in multiple tumour settings.

It is possible to decompose T-cell receptor mediated ERK signalling into two smaller sub-networks: (i) a receptor proximal signalling cascade with positive and negative feedback regulation, and (ii) the unidirectional MAP kinase (MAPK) cascade. We now demonstrate that inhibiting enzymes specific to each signalling sub-network produces a unique response in terms of ERK phosphorylation. To show this, we subjected activated T-lymphocytes to inhibitors targeting the two signalling sub-networks separately: a SRC inhibitor (Dasatinib) for the receptor proximal component, and a MEK inhibitor (PD325901) for the MAPK component (Fig. 3a). Importantly, the population-mean response of the cells to each inhibitor resulted in amplitude reduction and a trivial inhibition model (see Fig. 3d,e insets). However, going down to the single-cell resolution, the ppERK response to SRC inhibition (Dasatinib) resulted in an all-or-nothing response ('digital' inhibition, Fig. 3b,d). Conversely, application of a MEK inhibitor (PD325901) resulted in graded responses ('analogue' inhibition, Fig. 3c,e). SRC and MEK inhibitors exhibit markedly different modes of inhibition which do not rely on the exact chemical identity of the administered inhibitor but rather its role in the signalling cascade (Supplementary Notes 3.1 and 3.2).

We characterized the two modes of inhibition by fitting the distribution of ppERK amount per cell to a mixture of two Gaussians. The relevant statistics can be summarized by two parameters—$\alpha_{+}$ which represents the fraction of activated cells (Fig. 3f) and $\mu_{+}$ representing the mean ppERK levels among activated cells. We carried out this analysis for each dose of inhibitor. In Fig. 3g, we report that the MEK inhibitor operates solely upon the mean, $\mu_{+}$, of ppERK abundance among activated

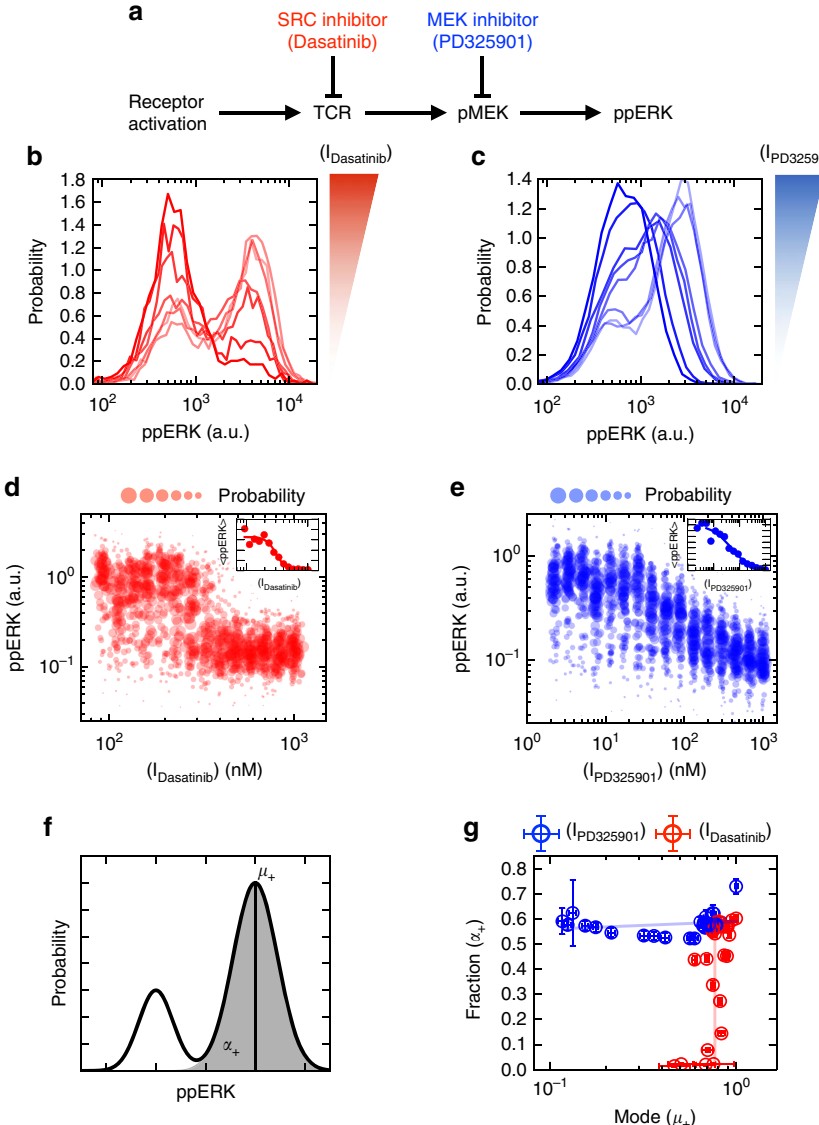

**Figure 3 | Specific modes of inhibition.** (**a**) The implicit model of inhibitor action. (**b**) Histograms of single-cell response to TCR stimulation and SRC inhibition with Dasatinib. (**c**) Histograms of single-cell response to TCR stimulation and MEK inhibition with PD325901. (**d**) Single-cell dose response of Dasatinib—the inset represents the mean response of all cells to a single dose of inhibitor. (**e**) Single-cell dose response of PD325901—the inset represent the mean response of all cells to a single dose of inhibitor. (**f**) Single-cell data were modelled as a mixture of Gaussian distributions. $\alpha_+$ parameterizes the fraction of activated cells, while $\mu_+$ parameterizes the average abundance of ppERK among activated cells. (**g**) The ($\mu_+$, $\alpha_+$) plane shows the orthogonal modes of inhibition (errorbars are standard error of mean from 100 samples of 500 T cells per dose of inhibitor chosen randomly and with replacement).

cells, which we define as analogue inhibition of ERK activation. In contrast, the SRC inhibitor operates solely upon the fraction of active cells, $\alpha_+$, which we define as digital inhibition. To summarize, by utilizing single-cell measurements, we were able to demonstrate that there exist two modes of inhibition in the MAPK signalling cascade, digital and analogue, each of which are associated with the sub-network the targeted kinase belongs. Each mode of inhibition can be associated with the unique inhibition of proximal and distal kinases respectively within the ERK cascade. We proceed to examine if this effect can be captured by the properties of the respective sub-networks and whether it maps to a functional output.

**Sub-network of targeted enzyme determines mode of inhibition.** We explored whether the two distinct modes of inhibition observed in Fig. 3 originate in the context of the targeted enzymes, that is, by the position of the enzyme undergoing

inhibition within the transduction cascade. For this, we developed a coarse-grained model which accounts for ERK phosphorylation downstream of SRC activation[24,36]. Our model explicitly incorporates measurable quantities (for example, abundance of CD8), control parameters (for example, antigen concentration), and the inhibitor targeted species[37], while incorporating uncontrollable or unmeasurable quantities into the phenomenological species 'activated SRC'. The graphical representation of our model (Fig. 4a) emphasizes the two sub-networks acting here. First, SRC* is controlled by competing positive and negative feedbacks that are abstractions of negative feedback of active SHP-1 phosphatase[24] and positive feedback associated with immune receptor signalling[36,38]. Second, MEK and ERK are activated upon formation of SRC* in an unidirectional manner, without feedback (see Supplementary Note 3.3 for experimental evidence for our unidirectional MAPK signalling assumption). These modelling components encompass

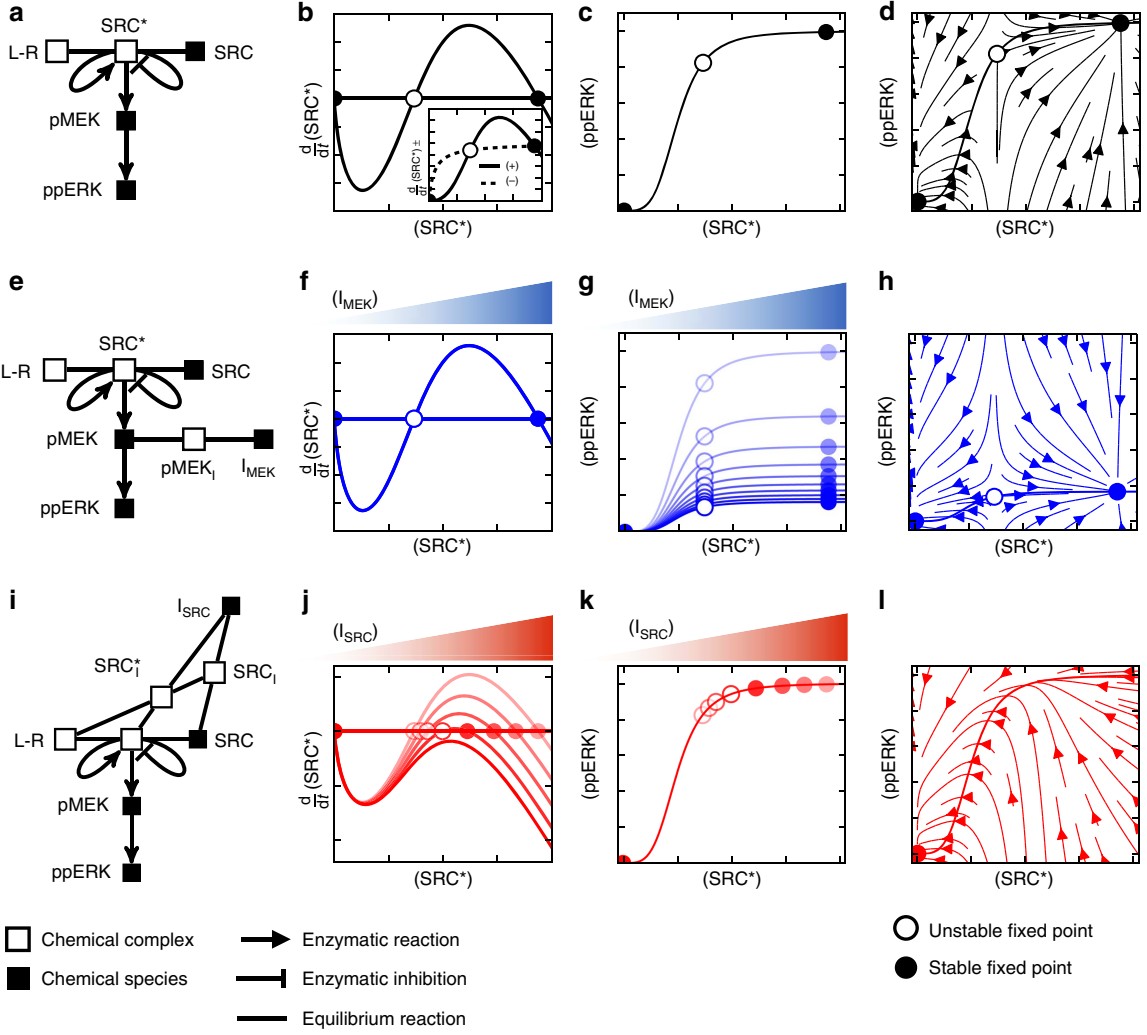

**Figure 4 | Sub-network context of the targeted enzyme determines response to inhibition.** (**a**) Model diagram of signalling network. (**b**) Phase plane of SRC* with respect to zero flux (horizonal line). The inset shows the behaviour of both the positive and negative model fluxes. (**c**) Functional response of ppERK to changes in SRC* abundance. (**d**) Instantaneous reaction velocities given ordered pairs of (SRC*, ppERK) shows the dynamic behaviour of the model system. (**e**) Model diagram for MEK inhibition of signalling. (**f,g**) Analogous to representations of model behaviour as (**b,c**) but for different doses of MEK inhibitor. (**h**) Instantaneous reaction velocities for maximal MEK inhibition. (**i**) Model diagram for SRC inhibition. (**j,k**) Analogous representations of SRC inhibition as **f** and **g** for MEK inhibition. (**l**) Instantaneous reaction velocities for maximal SRC inhibition show that the dynamics support a single fixed point at (SRC*, ppERK) = (SRC*$_{low}$, ppERKlow).

key features of ERK activation in the context of antigen activation in T lymphocytes.

To assess the properties of our competing feedback model we constructed phase diagrams of active SRC that demonstrate how varied quantities of active SRC map to ERK phosphorylation. Active SRC accumulates upon engagement of the activating ligand with its kinase-bearing receptor. This subsequently activates both positive and negative feedbacks driving further accumulation or extinction of SRC*. The dynamics of accumulation of SRC* (Supplementary Eq. 26) can be summarized in a phase diagram (Fig. 4b), that illustrates the influence of both feedbacks. The model parameters (Supplementary Table 2) are set so that the negative flux (that is, the change in time of SRC* levels) rises and saturates at lower levels of SRC* than the positive flux. This staggering of the positive and negative fluxes as a function of SRC* causes them be equal at three points in the phase diagram, that is, there are three 'steady states' or fixed points in our model. By plotting the net flux as a function of the active complex SRC*, we assessed the stability of the fixed points. The dynamics are such that SRC* always converges to the

extreme points SRC*$_{low}$ and SRC*$_{high}$ (stable fixed points), while diverging from the centre point SRC*$_{med}$ (unstable fixed point). Hence, our coarse-grained model encapsulates the bistability in SRC* formation.

We model ERK activation by assuming that the active complex SRC* triggers the enzymatic phosphorylation of MEK, which then phosphorylates ERK (Fig. 4c, Supplementary Equations 28 and 29, respectively). In Fig. 4d, we represent the dynamic trajectory of this signalling pathway for varied initial conditions: such a flow diagram illustrates the stability of the low and high states in the (SRC*, ppERK) plane and the instability of the intermediate point. Overall, our coarse-grained model of ERK activation upon ligand engagement generates two stable fixed points corresponding to either low or maximum ppERK, consistent with our experimental results.

Next, we tested whether our coarse-grained model can predict ppERK response to drug inhibition. Application of the MEK inhibitor to our model (Fig. 4e) supports our experimental observations, as MEK inhibition does not influence the bistability of the activated kinases SRC* (Fig. 4f). Increasing the MEK

inhibitor dose shows continuous reduction in the amplitude of ppERK response (Fig. 4g), without affecting the bistability in ppERK. The dynamic properties supporting the bistability in ppERK are preserved in the presence of the MEK inhibitor (Fig. 4h). Our model is validated with the experimental observation in that the MEK inhibitor only reduces the mean quantity of ppERK over the population of activated cells, that is, it inhibits the ERK pathway in an analogue manner.

Our model highlights that SRC is the kinase crucial for the bistability of the active complex SRC*, resulting in a signalling context fundamentally distinct to that of MEK. Inhibition of SRC reduces the positive flux which generates SRC* (Fig. 4i), and consequently reduces SRC* at the high fixed point. We find that increasing the dose of the SRC inhibitor decreases SRC* until, at a critical dose, the high fixed point and the unstable fixed point annihilate one another (Fig. 4j). Therefore, a dose of SRC inhibitor greater than the critical dose leaves the system with only a single fixed point, $SRC^* = SRC_{low}$ (Fig. 4j). Interestingly, despite the continuous reduction of the $SRC^*_{high}$ stable fixed point with increased dosage of SRC inhibitor, the quantity of ppERK remains essentially unchanged until the inhibitor is greater than the critical dose (Fig. 4k). For doses of SRC inhibitor beyond the critical dose the signalling network only supports a single low quantity ppERK (Fig. 4l). Hence, SRC inhibition results in a binary output that is identical to that observed in the data: our model is consistent with the digital nature of Dasatinib as a SRC inhibitor.

Our model assumes that interactions of molecular inhibitors with their target enzyme all act as noncompetitive inhibitors (consistent with *in vitro* characterization of these small molecules). Yet, despite these locally identical mechanisms of inhibition, the model successfully accounts for the two distinct modes of inhibition of ERK in our experimental findings. Thus, taken together, the experimental results and theoretical model demonstrate that enzymatic context is essential to understand and parameterize inhibitor function.

**Protein variability causes resilience to inhibition**. Using our coarse-grained model, we sought to explore how the endogenous variability of SRC abundance would diversify the response of individual cells to inhibition. Our model predicts that the effective quantity of SRC determines whether the (SRC, ppERK) phase diagram has a single or three fixed points—as a result it represents a bifurcation parameter (Supplementary Equation 32). By analogy, endogenous variation of SRC positions cells either above or below the critical threshold of SRC required for bistable signalling (Fig. 5a). We tested this hypothesis by correlating CD8 and ppERK of activated T-lymphocytes. In T lymphocytes, Lck—a SRC family kinase, is recruited together with CD8 to trigger response to antigen, therefore we treat CD8 abundance as a proxy for the effective abundance of SRC in individual cells. Indeed, measuring CD8 for a single dose of SRC inhibitor shows that cells with elevated quantities of CD8 are more likely to have ppERK signal (Fig. 5b), a result that is consistent with previous experimental and theoretical observations[37].

Extending this observation, our model suggests an interesting possibility: that variability of CD8 expression in single cells is sufficient to generate disparate sensitivities to drug inhibition. The bifurcation diagram for each drug dose, Fig. 5c, shows that the minimum quantity of SRC sufficient for the bistability, $SRC_c$, increases with increasing drug dose. Consequently, a cell with a higher abundance of SRC will be more tolerant to inhibition because of simple dosing of the effective abundance of available SRC (Fig. 5d)—requiring higher inhibitor dosage to experience any reduction in signalling. We confirmed this qualitative prediction by correlating the critical amount of CD8, labelled

$CD8_c$, with drug dose; the MEK inhibitor reduced ERK activation independently of the abundance of CD8 whereas higher concentrations of SRC inhibitor were required to inhibit (Fig. 5e,f).

**Mode of inhibition translates to proliferative response**. Having established the existence of distinct modes of inhibition of the ERK pathway, we conclude the results section of this communication by posing an important challenge to our finding: do these distinct modes of inhibition entail a functional ramification? Upon phosphorylation, ppERK migrates from the cytosol to the cell nucleus where it induces the expression of the immediate and early genes (IEGs, for example, cFOS). IEGs constitute a set of genes that facilitate cell cycle entry and cell division[39]. Hence it is reasonable to expect that inhibiting the ERK pathway will impact cell proliferation. But will cell proliferation, which happens on the scale of days, be sensitive to the different modes of inhibition, which happen on the scale of minutes? Our signalling results (Fig. 3) suggest the following hypothesis: that MEK inhibition, which produces intermediate levels of ppERK, will slow down induction of IEGs, and as a result would increase the time to cell division. In contrast, SRC inhibition, which reduces the fraction of cells getting activated, will reduce the number of cells entering cell cycle, without affecting the overall cell division in activated cells (cf Fig. 3).

We tested this hypothesis by quantifying the proliferation of T cells after 48 h of *in vitro* culture under concomitant antigen stimulation and drug exposure. We used flow cytometry to monitor cell activation and division by measuring cell size (FCS-A), the levels of the CD8 co-receptor on the surface of cells (proportional to fluorescence intensity), and the fluorescence of T cells that were tagged before activation with an amine-reactive fluorescent dye (CTV or CFSE, the dye gets diluted by two-fold at each cell division). Upon activation, T cells increase both their size and CD8 expression, providing a clear criterion (Fig. 6a) that separates inactive and active cell populations, whose numbers can be quantified as $N^-$ and $N^+$, respectively. Among the active fraction of cells we analyse the number of cells ($N_i^+$ for $i = \{0, 1, 2, 3, …\}$) undergoing $i$ divisions as measured by CTV or CFSE dilution (Fig. 6b). By computing both the mean number of divisions and the fraction of activated cells for each dose of each drug, we could plot the two hypothesized modes of long timescale inhibition.

Representing the data as fraction activated versus mean divisions, demonstrates that the disparate modes of inhibition for signal transduction map to the proliferative timescale (Fig. 6c). To be more explicit, we found that dosing of MEK inhibitor reduces the average number of divisions among activated cells, while the dominant feature of the SRC inhibition is the distinct reduction of the number of activated cells. This is not the exclusive feature observed in our data, since intermediate doses of SRC inhibitor do also reduce the mean divisions (possibly because of the unaccounted signalling transduction pathways dependent on TCR activation, for example, PI3K and AKT[40]). Crucially, application of MEK and SRC inhibitors shows grouping of the proliferation data when represented as fraction activated versus mean division number. We then found these results to be a general property of MEK and SRC inhibitors in our system by including the following: Bosutinib, PD325901, PP2, Trametinib and AZD6244; most these drugs are either presently clinically used or in various stages of clinical trials. Indeed, Fig. 6c shows an astonishing degree of agreement in-between the SRC inhibitors, in-between the MEK inhibitors, and at the same time, a very clear divergent behaviour of the two families. We conclude that our measurements support the hypothesis that the impact of MEK/SRC inhibition on cell proliferation recapitulate the two

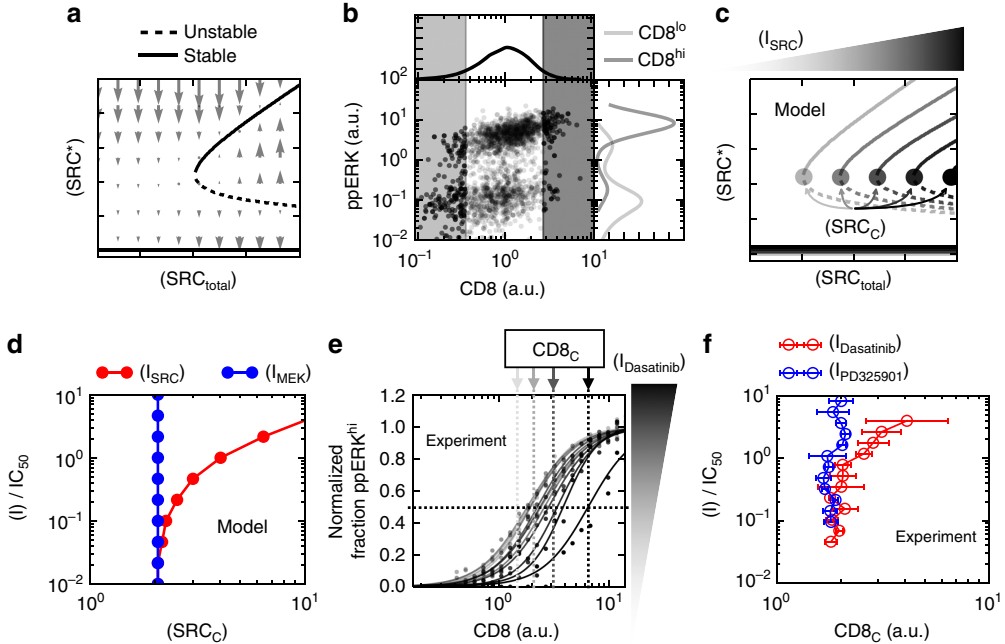

**Figure 5 | Protein variability tunes sensitivity of cells to inhibition.** (**a**) The abundance of $SRC_{total}$ controls the number of fixed points of the model, and consequently is a bifurcation parameter. (**b**) Flow cytometry measurements of T cells concomitantly labelled for CD8 (a proxy SRC) and ppERK (pT202, pY204) shows signalling dependence to endogenous expression of CD8. (**c**) Model predictions of $SRC_c$ dependence on SRC inhibitor dose. (**d**) Model predicts scaling of $SRC_c$ with increased SRC inhibitor doses. Intuitively, MEK inhibitor does not influence $SRC_c$. (**e**) CCVA of T cells concomitantly labelled for CD8 and ppERK (pT202, py204) treated with various doses of the SRC inhibitor Dasatinib. (**f**) Quantification of the half effective abundance of CD8 ($CD8_c$) required for ppERK activation in T cells treated with either MEK inhibitor (PD325901) or SRC inhibitor (Dasatinib). Errorbars quantify one standard deviation from the mean of experimental duplicate measurements.

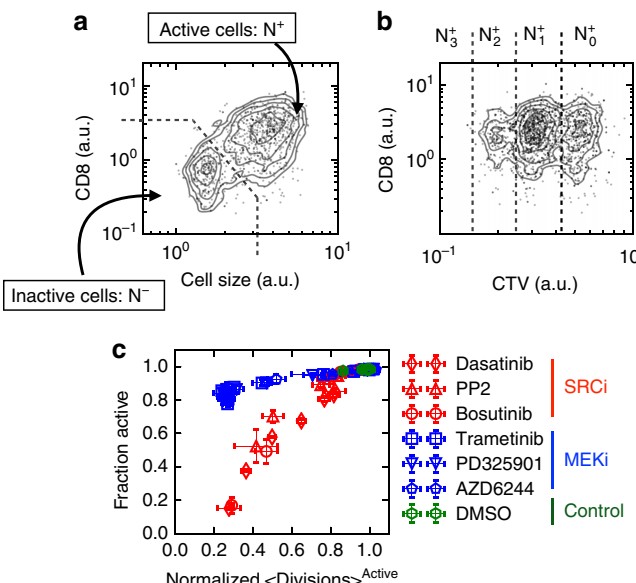

**Figure 6 | Short timescale modes of inhibition translate to long timescale proliferative response.** (**a**) Identification of activated T cells after 48 h of concomitant treatment with antigen and the respective inhibitor measured by flow cytometry. Upon activation, cells increase size, measured by forward scatter area (FSC-A), and upregulation of CD8. (**b**) CellTrace Violet (CTV) dilution identifies active cells that have divided $n$ times. (**c**) Quantification of fraction of active cells and mean number of divisions, among activated cells, for each inhibitor (see Supplementary Note 4 for derivation). Errorbars quantify one standard deviation from the mean of experimental duplicate measurements.

modes of inhibition that we documented with short-term signalling response in Fig. 3.

## Discussion

In this study we combined theoretical and experimental approaches to probe mechanisms of inhibition in signal transduction. We used single-cell phospho-profiling and CCVA[28] to characterize the biochemical details of small-molecule chemical inhibitors within living cells; such detail was so far limited to *in vitro* enzymatic assays. We uncovered a generic mechanism in which targeted enzyme inhibition manifests in two distinct patterns of inhibition, which we label 'digital' versus 'analogue'. Lastly, we probed the biological significance of these results by correlating short timescale signalling behaviour with unique modes of inhibition of cellular proliferation.

Using single-cell phospho-profiling and CCVA we were able to perform detailed and mechanistic characterization of cellular responses to targeted inhibition in primary cells. Specifically, we showed how to utilize CCVA to mechanistically characterize the biochemical interaction between the enzyme target and the inhibitor. We confirmed that AZD1480 is a potent noncompetitive, with respect to STAT5, inhibitor of JAK–STAT signalling in IL-2 stimulated primary T-lymphocytes (Fig. 2) and that pSTAT5 levels and drug efficacy depend on varied levels of endogenous STAT5. In addition, we demonstrated how the organization of reactions in biochemical networks in a more complex signalling cascade can determine markedly different cellular responses to inhibition (Figs 3 and 4).

Albeck *et al.*[27] already uncovered that inhibition of different enzymes manifest to digital or analogue signalling responses. Our contribution here is to propose a mechanistic model which attributes these disparate responses to the context of the targeted

enzyme. Essentially, the overall network response to inhibition is determined by the dynamic properties associated with the targeted enzymes' location in the larger biochemical network (Fig. 4). Furthermore, we show how our short timescale signalling behaviour translates to novel long timescale proliferative response to inhibition (Fig. 6). As a result, our method extends the characterization of inhibitors from the current state-of-the-art *in vitro* assays to primary single cells, and sheds light on the nonlinear signalling responses of the biochemical network structure being perturbed.

Building upon our initial findings, we extended our mechanistic models and used CCVA to demonstrate how cells utilize the endogenous variability of protein abundance to generate disparate responses to singular perturbations. In context to inhibition, we found that variation in enzyme substrate (STAT5) abundance established diverse signalling amplitudes and varied the sensitivity of single cells to inhibition (Figs 1 and 2). We then extended our mechanistic model of SRC inhibition and found that the variability of SRC expression operates on cells as a bifurcation parameter, which controls the number of possible steady states of the signalling network. As a result, cells that had elevated abundance of SRC were more tolerant to inhibition (Fig. 5). The extent in which these mechanisms of diversity provide resilience of populations to inhibition at longer timescales remains an open question. However, our findings are of practical importance: there are numerous examples of biological systems that utilize protein abundance to generate phenotypic variability, as noted in refs 41–45. Similarly, there exist abundant single-cell observations showing heterogeneous responses to inhibition[16,21,22].

Our method facilitates the extension of *in vitro* kinase assays to cellular systems and motivates a transition from phenomenological characterization of drug response at the single-cell level, into mechanistic and functional understanding. By using our combined approach of CCVA and development of mechanistic models to characterize drugs in primary cells, we were able to unravel fundamental chemical and biological processes. In particular, our method is especially useful when probing the functional consequences (on long timescales) of molecular perturbations (as experienced by cells on short timescales). When applied, we successfully showed how the SRC and MEK inhibitors cluster on two distinct curves, which are easy to interpret as distinct modes of inhibition. Since it is unlikely that the ERK pathway is the only cellular pathway exhibiting distinctly different modes, we expect that our method will prove useful in characterizing other inhibitor-pathway combinations, hopefully teasing out more novel modes of inhibition[46].

## Methods

**Mice and cells.** Primary splenocytes and lymphocytes were harvested from C57BL/6N (B6; Taconic Farms), B10A wild type (B10A; Taconic Farms), OT-1 TCR transgenic RAG2$^{-/-}$ (Taconic Farms), and 5C.C7 TCR transgenic RAG2$^{-/-}$ (Taconic Farms) mice and cultured up to 10 days. Mice were bred, maintained, and euthanized at Memorial Sloan Kettering Cancer Center (MSKCC) in compliance with our animal protocol. The animal protocol was reviewed and approved by the Institutional Animal Care and Use Committee (IACUC) of the Memorial Sloan Kettering Cancer Center (New York NY). The protocol number is 05-12-031 (last renewal data: 23rd December 2013). RMA-S TAP-deficient T-cell lymphoma cell line was used as APCs for signalling experiments[24].

**Antibodies and cell stains.** Cells were labelled with primary antibodies against doubly phosphorylated ERK 1/2 (pT202, pY204; clone E10; used at 1:300 dilution), phosphorylated MEK 1/2 (p-S221; clone 166F8; used at 1:100 dilution), phosphorylated STAT5 (p-Y694; clone C11C5; used at 1:200 dilution)—purchased from Cell Signaling Technology (Beverly, Massachusetts)—and polyclonal goat anti-STAT5 (catalogue number sc-835-G; used at 1:200 dilution) purchased from Santa Cruz Biotechnology (Santa Cruz, California). Secondary antibodies tagged with fluorescent molecules include PE conjugated donkey anti-mouse (catalogue number 715-116-151), APC conjugated donkey anti-mouse (catalogue number

715-136-151), FITC conjugated donkey anti-rabbit (catalogue number 711-097-003), and Alexa Fluor 647 conjugated donkey anti-goat (catalogue number 705-605-147) were all purchased from Jackson ImmunoResearch (West Grove, Pennsylvania; used at 1:200 dilution). In addition, Brilliant Violet 421 donkey anti-rabbit polyclonal antibody was purcahsed from BioLegend (San Diego, California; catalogue number 406410; used at 1:200 dilution). Surface markers CD8α (clone 53–6.7; used at 1:200 dilution) and CD4 (clone RM4-5; used at 1:300 dilution) tagged to fluorescent molecules were purchased from Tonbo biosciences (San Diego, California). Cell proliferation was measured by dilution of either CellTrace Violet (CTV) or Carboxyfluorescein *N*-succinimidyl ester (CFSE) proliferation kits purchased from Molecular Probes. Cell viability was assessed with Live/Dead Near-IR kit purchased from Molecular Probes.

**Small-molecule chemical inhibitors.** The SRC inhibitors PP2 and Bosutinib as well as the MEK inhibitor PD0325901 (PD325901) were purchased from Sigma-Aldrich. The MEK inhibitors Trametinib and AZD6244 were generous gifts from Neal Rosen (MSKCC). The JAK inhibitor AZD1480 and SRC inhibitor Dasatinib were purchased from Selleckchem.

**Additional reagents.** Supplemented RPMI-1640 media was prepared by MSKCC core media preparation facility and was used for all cell cultures and experiments. Media was supplemented with 10% fetal bovine serum, 10 µg ml$^{-1}$ penicillin and streptomycin, 2 mM glutamine, 10 mM HEPES (pH 7.0), 1 mM sodium pyruvate, 0.1 mM non-essential amino acids, and 50 µM β-mercaptoethanol. Cell were stimulated with with interleukin 2 (IL-2; eBioscience). TCR activating ligands K5 MCC peptide (K5): ANERADLIAYFKAATKF (T lymphocyte 5C.C7 agonist) and ovalbumin peptide SIINFEKL (T lymphocyte OT-1 agonist) were purchased from GenScript. Cells were chemical fixed and permeabilized following signalling experiments with 2% paraformaldehyde (PFA; Affymetrix) and 90% methanol (MeOH). Cells were stained with antibodies and suspended in FACS buffer for flow cytometry measurements. FACS buffer consists of 10% fetal bovine serum (MSKCC core media preparation facility) and 0.1% sodium azide in PBS. Ficoll-Paque PLUS (GE) was used to purify live cells in culture.

**Primary cell culture.** 5C.C7 and OT-1 primary cells were cultured *ex vivo* with peptide pulsed APCs from irradiated (3,000 RAD) B10A and B6 mice, respectively. APCs were pulsed overnight with 1 µM K5 peptide for 5C.C7 activation and 1 µM SIINFEKL for OT-1 activation prior to irradiation. Cells were purified by Ficoll-Paque gradient centrifugation and given exogenous IL-2 (1 nM) every other day. All cells were cultured at 37 °C and 5% CO$_2$ in supplemented RPMI and used for experiments within 7 days of activation.

**Single-cell inhibition of signal transduction assay.** The pSTAT5 response to JAK inhibition was measured using primary 5C.C7 derived T lymphocytes. Cells were aliquoted in 96-well v-bottom plates with exogenous IL-2 (working dilution 2 nM) for 10 min and kept at 37 °C. The cells were then treated the JAK inhibitor AZD1480 for 15 min at 37 °C. Followed by 15 min of fixing in 2% PFA on ice. The cells were then permeabilized in 90% MeOH and stored at −20 °C until staining for flow cytometry.

The ppERK response to SRC and MEK inhibition was measured using primary OT-1 T-lymphocytes activated with RMA-S APCs. RMA-S cells were suspended in culture with 1 nM SIINFEKL peptide for 2 h at 37 °C, 5% CO$_2$, and on a rotator to guarantee mixing. During this time we labelled OT-1 cells with an amine-reactive dye, CTV, according to the manufacture's protocol (Molecular Probes). This fluorescent tag was used to identify OT-1 cells *in silico*. We rested the OT-1 cells one hour after CTV staining, and then distributed them in a 96-well v-bottom plate. Each well was given various doses of SRC inhibitor and MEK inhibitor and kept at 37 °C for 5 min. Following the 5 min exposure to the inhibitors, we added the peptide pulsed RMA-S (10 RMA-S to 1 OT-1 T cell) and pelleted by centrifugation for 10 s at 460 rcf at room temperature. This step guaranteed that both cell types, OT-1 and RMA-S, came into contact. The cells were allowed to activate for 10 min, followed by fixing on ice in 2% PFA, and then permeabilized and stored in 90% MeOH at −20 °C.

**Proliferation assay.** The proliferative response of OT-1 T-lymphocytes to SRC and MEK inhibitors was measured by the dilution of the amine-reactive dyes CTV or CFSE. Splenocytes from B6 mice were used as APCs. Once harvested, the APCs, were given exogenous SIINFEKL peptide (1 nM) for 2 h and kept at 37 °C, in 5% CO$_2$, and placed on a rotator to guarantee mixing. During this time, lymphocytes and splenocytes were harvested from an OT-1 mouse, and labelled with either CTV or CFSE according to the manufacture's protocol (Molecular Probes). After the 2 h of incubation, the B6 splenocytes were irradiated with 3,000 rad. Irradiated B6 splenocytes and CTV or CFSE stained OT-1 lymphocytes and splenocytes were mixed, 10 B6 derived cells per OT-1 derived cell, in sterile 96-well v-bottom plates. The inhibitors were then administrated and the plates were kept at 37 °C and in 5% CO$_2$ for 48 h. After the 48 h, cells were labelled with a fixable Live/Dead stain according to manufacture's protocol (Molecular Probes), fixed in 2% PFA, and suspended in 90% MeOH at −20 °C until staining for flow cytometry.

**Data analysis.** Flow cytometry measurements were compensated and gated using FlowJo software. All other data analysis was performed using the scientific python software suite (SciPy), figures were produced in matplotlib[47], and Gaussian mixture modelling performed using scikit-learn[48].

**Data availability.** The data that supports the findings of this study are available upon request from the corresponding author.

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

## Acknowledgements

This research was funded by NIH U54 CA148967, the Geoffrey Beene Cancer Center at MSKCC and the intramural research program of the National Cancer Institute. A.E. is supported by the Human Frontier Science Program grant LT000123/2014. Furthermore, we thank Neal Rosen for commentary and for generously sharing inhibitors. We would also like to thank Jacqueline Bromberg for her valuable comments and discussions.

## Author contributions

R.V. developed models; R.V. and G.A.-B. designed experiments; R.V. and A.E. Analysed data; R.V., A.E. and G.A.-B. performed experiments and wrote the manuscript.

## Additional information

**Competing financial interests:** The authors declare no competing financial interests.

