## [Peer Review File · Nature Communications]

Reviewer #1 (Remarks to the Author)

A. Summary of the key results

The study by Vogel, Erez and Altan-Bonnet shows how computational models can account for distinct, experimentally observed modes of signaling inhibition in T lymphocytes. First, they show by model selection based on flow cytometry data and cell-to-cell variability analysis of IL2-induced STAT5 phosphorylation that the JAK inhibitor AZD1480 acts by non-competitive binding to the kinase. They reveal by the same approach qualitatively different effects of the SRC inhibitor Dasatinib and the MEK inhibitor PD325901 on ERK phosphorylation. The model predicts that higher SRC abundance renders T lymphocytes more tolerant to SRC inhibition but not MEK inhibition, which is experimentally validated. These two modes of inhibition were found to be translated into the proliferation response of T lymphocytes after 48 hours for each 3 different inhibitors of SRC or MEK, respectively. SRC inhibition was found to reduce the fraction of actively proliferating cells while MEK inhibition reduced the number of divisions of actively proliferating cells.

B. Originality and interest: if not novel, please give references

It is very interesting that the two proposed modes of short-term signaling inhibition translate to the long-term proliferation response. Despite the fact that it is intuitive, such a direct connection has not been shown before.

C. Data & methodology: validity of approach, quality of data, quality of presentation

Analyzing cell-to-cell variability is a valid approach to disentangle modes of inhibition in a signaling pathway, given that heterogeneity in protein abundance may compensate for the inhibitor's effects in individual cells (by stoichiometric buffering).

The quality of the data seems appropriate. The authors may consider showing individual data points for the two conducted experiments instead of mean and standard deviation. It is unclear why the y axis of the inlay in Figure 1D was rescaled compared to Figure 1B.

It needs to be clearly stated in the main text at which time point signaling measurements were performed and why.

D. Appropriate use of statistics and treatment of uncertainties

A coefficient of determination is given to indicate how STAT5 relates to pSTAT5 for treatment with JAK inhibitor. Confidence intervals could be provided for the linear correlation. The authors should also show the prediction for the other uncompetitive and the competitive model in Fig. 2E. For model selection, the authors have to provide a statistical measure that takes the number of model parameters into consideration.

The authors speculate that diverse levels of pSTAT5 may be caused by unique internal parameters. They should specify which parameters they mean.

E. Conclusions: robustness, validity, reliability

The all-or-nothing feature of the model crucially relies on the negative and positive auto-feedback from/to SRC* in the model as shown in Figure 4A, E and I. It needs to be clarified on which assumptions these feedbacks have been based. With respect to phase diagrams, the authors should avoid referring to "dynamics" as this would imply any information with respect to time that cannot be drawn from such analysis. For the two modes of the inhibitors, the authors only reproduce two qualitative phenomena. They should comment on quantitative aspects such as the threshold for ppERK, the abundance of SRCc or CD8, and the inhibitor doses proposed in the model prediction (Figure 5D, y axis) that are in discrepancy with those experimentally measured (Figure 5E, y axis). The authors should further discuss how the two modes of inhibition mechanistically could be linked to the proliferation response.

F. Suggested improvements: experiments, data for possible revision

The authors claim the MAP kinase cascade to be unidirectional without a common negative feedback. In other systems, multiple negative feedback from activated ERK to upstream components have been shown. Frequently, if MEK was inhibited, the amount of phosphorylated

MEK was increased due to these negative feedback. To show that the assumption of a unidirectional MAPK pathway hold true in T lymphocytes, experimental data showing that upon MEK inhibition ppMEK levels are not increased in T lymphocytes is required.

G. References: appropriate credit to previous work?

The proposed mechanism of inhibition of the JAK inhibitor AZD1480 was claimed to be supported by the literature even though no reference was given. A link between signaling data and proliferation in response to inhibitor treatment has been described before: Kirouac et al. (2013) *Sci. Signal.* 6: ra68.

H. Clarity and context: lucidity of abstract/summary, appropriateness of abstract, introduction and conclusions

The authors should rather pronounce the inference of phenotypic response from signal perturbations. In this work, a snap shot from an early signaling event is connected to a snap shot of a late proliferation observation. However, it was not further investigated how informative these time points have been.

The authors should not mention cell lines as models for physiological settings especially as they have used primary material, and cell lines often represent a transformed and therefore non-physiological context (in terms of protein abundance and proliferation). Minor point: Please note that the Supplementary Information has a different title than the main text.

Reviewer #2 (Remarks to the Author)

In this manuscript, Vogel et al. elucidate principles and mechanisms of single cell responses to small molecule inhibitors. They use a data analysis technique called cell-to-cell variability analysis (CCVA) that was previously developed by the same group (Cotari et al., 2013) to quantify the relationships linking heterogeneity in protein abundance and regulatory function in single cells. The current manuscript expands significantly on their previous work by disentangling mechanisms of drug action from single-cell flow data. They specifically demonstrate the response amplitude and IC50 for AZD1480 scales in proportion with the protein abundance of STAT5 in single cells using CCVA, and validate a non-competitive model of inhibition by weighing mechanistic simulations against single-cell data. In addition to characterization of systems dynamics, they subsequently show digital and analog responses to inhibitors that target either proximal or distal subnetworks within the ERK signaling pathway. The abundance of SRC is identified as a critical threshold essential to bistability of the digital response in single-cells. They conclude with a functional study of cellular responses over longer timescales and demonstrate that analog and digital inhibition can selectively target either proliferation rate or proliferation rate in addition to fraction of activated T cells.

Overall, the main strengths of this manuscript are not necessarily the specific mechanisms of STAT5, SRC and Mek inhibition, but the process of parsing single cell data to generally understand mechanisms of drug inhibition and predict functional consequences of cell-to-cell heterogeneity. This demonstration is mathematically detailed, yet elegantly presented to be accessible to a broad readership. Although the manuscript is well written, there are several minor issues of clarity that should be addressed before publication.

- References to supplementary information are non-specific throughout the manuscript, and should be corrected to refer to the appropriate supplementary section, figure, or formula. For example, I spent far longer than necessary to find in the supplement the linearization transform used in fig 2D.

-References to panels of figure 3 in the main text are incorrect.

- Because Nature Communications is a multidisciplinary journal, several concept should be more carefully defined or explained in the manuscript. One example are the distinctions between un-

competitive, non-competitive, and competitive models for the mode of action of AZD1480. Another example is the basic principle and use of CCVA in understanding the relationships between protein abundance, regulatory function, and sensitivity to stimuli.

- It's intriguing that different inhibitors of proximal and distal signaling (SRC and MEK) can switch from having digital to analog characteristics. However, the analysis very much depends on the assumption that the cell distributions for both inhibitor classes can be equally well-fit by a 2-component Gaussian mixture model. I would be even more confident in their interpretation if the choice of a 2-component model were validated, using BIC (or another criterion) to determine the number of components that best describe each of the inhibitor classes.

- the chosen mode of action for I_{src} and I_{mek} on the subnetwork in figure 4 is not discussed or referenced. In the context of competitive, non-, and un-competitive modes of inhibition, how were the models for I_{src} and I_{mek} chosen? i.e. I_{src} can bind to the engaged receptor complex, but I_{mek} cannot bind with p_{mek} . Can other models of I_{src} and I_{mek} inhibition also generate digital and analog responses, or is this property unique to the assumed modes of inhibition?

- zero value for phase diagram should be marked, or identified as the flat line in the figure legend for 4B, F and J. Similarly for Fig. 5 A & C, also note that the low branch in Fig. 5A is nearly indistinguishable from the x-axis.

- they present a simple coarse grained model for ERK activation that recapitulates digital and analog inhibition. The bistable response of Src, whereby the stable fixed points are either SRC*low or SRC*high, result from a subnetwork architecture with positive and negative feedback. This subnetwork architecture seems essential for the ultrasensitive-like response and digital inhibition (i.e. the switch does not partially fire so the inhibitor's influence is all or none), yet the biological rationale for these mechanisms are not discussed in the text. These should be elaborated with references.

-fonts for the positive and negative flux inset of figure 4b are very small when printed.

-on page 7, end of paragraph 2 "Overall, our coarse grained... corresponding to either zero or maximum ppERK, consistent with our experimental results." - There is no data demonstrating zero ppERK. This should either be referred to as 'low' ppERK, or control experiments should be included that demonstrate the inactive ppERK peak to the left side of the distribution in Fig. 3F corresponds to zero ppERK (i.e. non-specific distribution).

- Description of errorbars for figure 5 & 6

-For Figure 6c it should be more clearly described in the main text that the axis for "normalized mean number of divisions" is derived from only the active cell population.

Reviewer 1 (Remarks to the Author):

A. Summary of the key results The study by Vogel, Erez and Altan-Bonnet shows how computational models can account for distinct, experimentally observed modes of signaling inhibition in T lymphocytes. First, they show by model selection based on flow cytometry data and cell-to-cell variability analysis of IL2-induced STAT5 phosphorylation that the JAK inhibitor AZD1480 acts by non-competitive binding to the kinase. They reveal by the same approach qualitatively different effects of the SRC inhibitor Dasatinib and the MEK inhibitor PD325901 on ERK phosphorylation. The model predicts that higher SRC abundance renders T lymphocytes more tolerant to SRC inhibition but not MEK inhibition, which is experimentally validated. These two modes of inhibition were found to be translated into the proliferation response of T lymphocytes after 48 hours for each 3 different inhibitors of SRC or MEK, respectively. SRC inhibition was found to reduce the fraction of actively proliferating cells while MEK inhibition reduced the number of divisions of actively proliferating cells.

B. Originality and interest: if not novel, please give references It is very interesting that the two proposed modes of short-term signaling inhibition translate to the long-term proliferation response. Despite the fact that it is intuitive, such a direct connection has not been shown before.

We thank the referee for appreciating the novelty of our findings.

C. Data & methodology: validity of approach, quality of data, quality of presentation Analyzing cell-to-cell variability is a valid approach to disentangle modes of inhibition in a signaling pathway, given that heterogeneity in protein abundance may compensate for the inhibitor's effects in individual cells (by stoichiometric buffering). The quality of the data seems appropriate. The authors may consider showing individual data points for the two conducted experiments instead of mean and standard deviation. It is unclear why the y axis of the inset in Figure 1D was rescaled compared to Figure 1B.

The inset in Figure 1D is in linear scale, because we model the changes of pSTAT5 according the variation in STAT5 in linear scale.

It needs to be clearly stated in the main text at which time point signaling measurements were performed and why.

Thank you for pointing out this important detail that we omitted in our original manuscript. We have added text to the manuscript stating that all measurements were performed after 15 minutes of inhibition. We chose this time-point after checking that the dynamics of pSTAT5 signaling reached a stationary value with characteristic timescales of 7 minutes: we included a new supplementary section (Page 2 of Supplementary Material) and figure establishing this timescale experimentally and demonstrating the appropriateness of our steady-state approximation (our new Supplementary Figure S1). Please note that we chose this short activation timespan because it is long enough to reach steady-state, yet sufficiently short that we can safely assume that gene regulatory effects remain negligible and focus our modeling on signaling responses only.

D. Appropriate use of statistics and treatment of uncertainties A coefficient of determination is given to indicate how STAT5 relates to pSTAT5 for treatment with JAK inhibitor. Confidence intervals could be provided for the linear correlation. The authors should also show the prediction for the other uncompetitive and the competitive model in Fig. 2E.

Thank you for your attention to this detail - we have removed the coefficient of determination from Figure 2D, because this value does not represent the statistics used for our model selection. We would like to point out that the line in Figure 2D is not a fit, but a guide to reader's eye, showing that the transformed data follows a simple line, $y = x$.

For model selection, the authors have to provide a statistical measure that takes the number of model parameters into consideration.

We have added a discussion to this point in Section 1.5, "Model selection", in the supplemental materials (Page 8). Statistical measures that incorporate the model complexity, or number of parameters (e.g. BIC or AIC) are essential when the number of parameters among the compared models are different. However, all

the models that we consider have the same number of parameters, making obsolete the use of these otherwise important tools.

The authors speculate that diverse levels of pSTAT5 may be caused by unique internal parameters. They should specify which parameters they mean.

We have expanded our logical flow (Edits line 124-126 of main text). The internal parameters we were referring to (first implicitly then explicitly) referenced is number of STAT5 proteins per cell.

E. Conclusions: robustness, validity, reliability The all-or-nothing feature of the model crucially relies on the negative and positive auto-feedback from/to SRC* in the model as shown in Figure 4A, E and I. It needs to be clarified on which assumptions these feedbacks have been based.

We included the feedbacks into our model based upon previous documentation of their biological relevance. Specifically, the negative feedback was motivated by the previously documented dynamics of SHP-1 [Altan-Bonnet and Germain, *PLoS Biology* (2005) 3(11): e356] and the positive feedbacks from general immune receptor activation [Das et al., *Cell* (2009) 136(2):337-51], and [Mukherjee et al. *Science Signaling* (2013) 6(256):ra1]. Please refer to lines 195-197 to see how we specifically integrated these details and references into the main text.

With respect to phase diagrams, the authors should avoid referring to "dynamics" as this would imply any information with respect to time that cannot be drawn from such analysis.

We respectfully disagree in the appropriateness of the word dynamics. Our analysis is predicated on the phenomena producing our experimental observations is derived from a nonlinear dynamical process, namely bistability in the dynamics of SRC activation. Furthermore, the tools of our analysis are those explicitly used to study complex dynamical systems. Lastly, we feel that our dynamic approach is experimentally justified because of the many papers documenting both positive and negative feedbacks in early TCR signaling events [Das et al., *Cell* (2009) 136(2):337-51], [Altan-Bonnet and Germain, *PLoS Biology* (2005) 3(11): e356], and similarly in B cells [Mukherjee et al. *Science Signaling* (2013) 6(256):ra1].

For the two modes of the inhibitors, the authors only reproduce two qualitative phenomena. They should comment on quantitative aspects such as the threshold for ppERK, the abundance of SRCc or CD8, and the inhibitor doses proposed in the model prediction (Figure 5D, y axis) that are in discrepancy with those experimentally measured (Figure 5E, y axis).

Indeed, the coarsening of our model description of early TCR events precludes a direct quantitative assessment of the variable sensitivity of cells to SRC inhibition. We thank the referee for identifying a potential point of confusion for the reader and have specified that our prediction was qualitatively validated in the main text (line 262). We hope that the referee and the reader will appreciate how our coarse-grained model was used to generate new hypothesis, and not to perfectly reproduce the quantitative scaling between SRCc and IC₅₀. We find that our coarse-grained approach yields very clear insights as well as qualitative predictions that we validated experimentally.

The authors should further discuss how the two modes of inhibition mechanistically could be linked to the proliferation response.

We feel that a full dissection of the potential mechanisms linking our signaling observations to proliferation responses would be beyond the scope of our study. The main text does include a discussion on the importance of ppERK's translocation to the cell nucleus, the induction of the Immediate Early Genes (IEGs), and the importance of IEGs for cell cycle entry and cell division: this directly links the analog/digital dichotomy observed in signaling with later functional readout (main text lines 270-278).

F. Suggested improvements: experiments, data for possible revision The authors claim the MAP kinase cascade to be unidirectional without a common negative feedback. In other systems, multiple negative feedback from activated ERK to upstream components have been shown. Frequently, if MEK was inhibited, the amount of phosphorylated MEK was increased due to these negative feedback. To show that the assumption of a unidirectional MAPK pathway hold true in T lymphocytes, experimental data showing that upon MEK

inhibition ppMEK levels are not increased in T lymphocytes is required.

We thank the reviewer for bringing this point to our attention. We had presented experimental results for the proposed measurement in Supplemental Material Section 3.2, Figure S12A-B. However, we feel that this point is essential to convey clearly to readers, and as a result, we have included a new supplemental section (Section 3.3) to specifically address the issue of pMEK steady state. In this section we present analysis of phosphorylated MEK (pS221) for MEK inhibited cells (analysis is of data presented in Figure S12A-B). Our experimental data and analysis clearly supports our coarse-grained approximation of unidirectionality in MAPK pathway, as probed in this paper (e.g. short timescale).

G. References: appropriate credit to previous work? The proposed mechanism of inhibition of the JAK inhibitor AZD1480 was claimed to be supported by the literature even though no reference was given. A link between signaling data and proliferation in response to inhibitor treatment has been described before [Kirouac et al. (2013) *Sci. Signal.* 6: ra68].

Thank you for these suggestions. We had included the reference (line 147), [Hedvat et al. *Cancer Cell* (2009) 16(6):487-97], in which the competitive mode of JAK inhibition for AZD1480 was established. The link between signaling data and proliferation in response to inhibitor treatment is indeed a classical issue for researchers working on developing drug inhibitors.

H. Clarity and context: lucidity of abstract/summary, appropriateness of abstract, introduction and conclusions The authors should rather pronounce the inference of phenotypic response from signal perturbations. In this work, a snap shot from an early signaling event is connected to a snap shot of a late proliferation observation. However, it was not further investigated how informative these time points have been.

We have addressed the relevance of the 15 min snap shot for the signaling event in our reply to comment C above. Regarding the 48hr snap-shot for the proliferation observation, we note that it is well established in the literature that the 48hr time-point is a significant time to study proliferation: it is late enough for a clear proliferative measurement yet early enough not to suffer from issues such as depletion of media or termination of the T-cell proliferative program [Marchingo et al. *Science* (2014) 346(6213):1123-7]. We are in the process of researching the dynamics of proliferation across several snap-shots ranging from 8 to 60 hours. All our results are consistent with the conclusions from the 48hr snap-shot presented in this work, yet, the full dynamics of cell proliferation remain too complex and well beyond the scope of this work.

The authors should not mention cell lines as models for physiological settings especially as they have used primary material, and cell lines often represent a transformed and therefore non-physiological context (in terms of protein abundance and proliferation).

We respectfully disagree. We mentioned cell lines in the introduction as they remain the go-to models for many pharmacological studies of drug inhibition. We agree with the Referee that cell lines are not ideal models (hence our work on primary lymphocytes). However, we believe that our method is general and applicable to all the readership of *Nature Communications*.

Minor point: Please note that the Supplementary Information has a different title than the main text.

We corrected this mistake.

Reviewer 2 (Remarks to the Author):

In this manuscript, Vogel et al. elucidate principles and mechanisms of single cell responses to small molecule inhibitors. They use a data analysis technique called cell-to-cell variability analysis (CCVA) that was previously developed by the same group (Cotari et al., 2013) to quantify the relationships linking heterogeneity in protein abundance and regulatory function in single cells. The current manuscript expands significantly on their previous work by disentangling mechanisms of drug action from single-cell flow data. They specifically demonstrate the response amplitude and IC50 for AZD1480 scales in proportion with the protein abundance of STAT5 in single cells using CCVA, and validate a non-competitive model of inhibition by weighing mechanistic simulations against single-cell data. In addition to characterization of systems dynamics, they subsequently show digital and analog responses to inhibitors that target either proximal or distal subnetworks within the ERK signaling pathway. The abundance of SRC is identified as a critical threshold essential to bistability of the digital response in single-cells. They conclude with a functional study of cellular responses over longer timescales and demonstrate that analog and digital inhibition can selectively target either proliferation rate or proliferation rate in addition to fraction of activated T cells.

Overall, the main strengths of this manuscript are not necessarily the specific mechanisms of STAT5, SRC and Mek inhibition, but the process of parsing single cell data to generally understand mechanisms of drug inhibition and predict functional consequences of cell-to-cell heterogeneity. This demonstration is mathematically detailed, yet elegantly presented to be accessible to a broad readership. Although the manuscript is well written, there are several minor issues of clarity that should be addressed before publication.

- References to supplementary information are non-specific throughout the manuscript, and should be corrected to refer to the appropriate supplementary section, figure, or formula. For example, I spent far longer than necessary to find in the supplement the linearization transform used in fig 2D.

We thank the referee for this suggestion and have made our references to the Supplementary Material more specific by adding sections and/or formula numbers where necessary.

-References to panels of figure 3 in the main text are incorrect.

Thank you for addressing these mistakes, we have corrected the errors.

- Because Nature Communications is a multidisciplinary journal, several concept should be more carefully defined or explained in the manuscript. One example are the distinctions between un-competitive, non-competitive, and competitive models for the mode of action of AZD1480. Another example is the basic principle and use of CCVA in understanding the relationships between protein abundance, regulatory function, and sensitivity to stimuli.

We added a more complete description of the varied modes of drug inhibition (lines 141-144 main text) and our CCVA method (lines 128-130 main text). We would like to point out that we are referencing one review [Cotari et al., *Current Opinion in Biotechnology* (2013) 24(4):760-6] where additional details can be found.

- It's intriguing that different inhibitors of proximal and distal signaling (SRC and MEK) can switch from having digital to analog characteristics. However, the analysis very much depends on the assumption that the cell distributions for both inhibitor classes can be equally well-fit by a 2-component Gaussian mixture model. I would be even more confident in their interpretation if the choice of a 2-component model were validated, using BIC (or another criterion) to determine the number of components that best describe each of the inhibitor classes.

We agree with the Referee that our analysis depends on the choice of a 2-component mixture as opposed to 3- or higher number-component mixture. To test this point we computed the Bayesian Information Criteria (BIC),

$$\text{BIC} = \underbrace{-2\log(\mathcal{L})}_{\text{Quality of fit}} + \underbrace{d\log(N)}_{\text{Parsimony Constraint}} \quad (1)$$

to find the optimal number of Gaussian mixtures. Here \mathcal{L} is the likelihood of the data, N is the number of data points and d is the number of parameters of the model. We compute BIC for mixtures of either

1, 2, 3, ..., 8 bivariate Gaussian distributions of log-scale (CD8, ppERK) single cell data for each dose of inhibitor (below). Each set of figures represents the calculation for a unique experimental replicate. Lastly, we show the optimal number of components, 'K', for each dose of PD325901 (MEKi) and Dasatinib (SRCi). We find that, on average over all experiments, $K_{\text{optimum}} < 2.5$: this motivated our modeling of the system as a two-component system.

Replicate 1

Replicate 2

MEK Inhibitor, PD325901 : Each BIC value represents an average BIC over 50 bootstraps with each bootstrap consisting of 1000 cell samples selected randomly with replacement. We normalize BIC(K) by BIC(K = 2) to emphasize that the two component model is frequently the optimum model.

Replicate 1

Replicate 2

SRC Inhibitor, Dasatinib : Each BIC value represents an average BIC over 50 bootstraps with each bootstrap consisting of 1000 cell samples selected randomly with replacement. We normalize BIC(K) by BIC(K = 2) to emphasize that the two component model is frequently the optimum model.

A

B

The optimal number of Gaussian mixture components. The average K_{optimum} and error bars representing one standard deviation over biological triplicates for (A) PD325901 (MEKi) and (B) Dasatinib (SRCi).

- the chosen mode of action for Isrc and Imek on the subnetwork in figure 4 is not discussed or referenced. In the context of competitive, non-, and un-competitive modes of inhibition, how were the models for Isrc and Imek chosen? i.e. Isrc can bind to the engaged receptor complex, but Imek cannot bind with pmek.

In short, our choice of mechanism is inspired by the biology of these processes.

SRC Inhibition: The underlying biology that produces our phenomenological SRC* is independent to the kinase activity Lck, the SRC family kinase of the system. Explicitly, SRC*, represents the physical coupling of the antigen carrying Major histocompatibility complex (pMHC) from the antigen presenting cell, T cell receptor (TCR), and Lck bound to CD8. At no point is the coupling of pMHC, TCR, and CD8-Lck dependent on the enzymatic activity of Lck. Consequently, we conjecture that the inhibitor does not effect the ability of the complex to form, making the noncompetitive mechanism the most rational choice.

MEK Inhibition: We thank the referee for the attention to this detail. Indeed our naming convention for pMEK bound to the MEK inhibitor in Figure 4 of the main text, and the model equations in the supplemental material were unclear. We have changed them accordingly to specifically indicate that MEK inhibitors bind to pMEK.

- Can other models of Isrc and Imek inhibition also generate digital and analog responses, or is this property unique to the assumed modes of inhibition?

It is indeed our understanding that the digital response of ppERK signaling to the SRC inhibitor is dependent on the nonlinearity of the pathway and not on the exact mechanism of action of the drugs. This becomes evident if we consider the model equations (Eqs. S23, S24 and S26), and the intuition built in the JAK-STAT inhibitor analysis. We see that the different mechanisms of drug action change how we incorporate the inhibitor equations to the total SRC mass conservation equation. This, in turns, influences the functional dependence of free [SRC] on [I] (Eq. S24). However, regardless of the functional form of Eq. S24, the influence of the inhibitor on the coarse-grained dynamic equation for SRC* is incorporated identically (Eq. S25). Accordingly, addition of any SRC inhibitor will consequently reduce the forward flux of SRC*, and because of the structure of the signaling pathway will always result in digital inhibition.

One should consider the mechanism of inhibition to study how the endogenous variability of protein influences a cell's sensitivity to the inhibitor. To understand this point let's re-examine the mechanisms of inhibitor action from the JAK-STAT study (Eqs. S6, S12, and S17) in the table below.

Inhibition mode	Mechanism of action	Equation
Noncompetitive	Reduces signaling response amplitude	S6
Uncompetitive	Reduces signaling amplitude and half effective concentration of substrate	S12
Competitive	Increases the half effective concentration of substrate	S17

The different parameters that each inhibitor modulates is indicative of how cells can evade inhibition. Take, for example, that both the noncompetitive and uncompetitive modes of inhibition reduce the maximal signaling amplitude. Biologically, the signaling amplitude is proportional to the abundance of the enzyme per cell. Consequently, a cell can evade inhibition by increasing the abundance of the inhibitor's target enzyme, a fact show in in Figure 5 of the main text. In contrast, the competitive inhibitor increases the half effective concentration of *substrate* required for maximal signaling response. Therefore cells can evade inhibition by increasing the abundance of the target enzyme's *substrate*. We hope that this example highlights the utility in understanding the mode of inhibition in context to the digital signaling response, and the depth of understanding that our modeling approach provides.

- zero value for phase diagram should be marked, or identified as the flat line in the figure legend for 4B, F and J. Similarly for Fig. 5 A & C, also note that the low branch in Fig. 5A is nearly indistinguishable from the x-axis.

Thank you for identifying this ambiguity in our manuscript. We have addressed this issue by specifying the identity of zero flux lines in the figure captions (Fig 4B,F,J), and modifying the y axis limits in Figures 5A,C.

- they present a simple coarse grained model for ERK activation that recapitulates digital and analog inhibition. The bistable response of Src, whereby the stable fixed points are either SRC*low or SRC*high, result from a subnetwork architecture with positive and negative feedback. This subnetwork architecture seems essential for the ultrasensitive-like response and digital inhibition (i.e. the switch does not partially fire so the inhibitor's influence is all or none), yet the biological rationale for these mechanisms are not

discussed in the text. These should be elaborated with references.

Please see our answer to a similar question from Reviewer 1.

-fonts for the positive and negative flux inset of figure 4b are very small when printed.

Thank you, we have enlarged the fonts in this figure.

-on page 7, end of paragraph 2 "Overall, our coarse grained... corresponding to either zero or maximum ppERK, consistent with our experimental results." - There is no data demonstrating zero ppERK. This should either be referred to as 'low' ppERK, or control experiments should be included that demonstrate the inactive ppERK peak to the left side of the distribution in Fig. 3F corresponds to zero ppERK (i.e. non-specific distribution).

We have altered our text to emphasize a low SRC* and ppERK levels, as recommended.

- Description of errorbars for figure 5 & 6

We thank the referee for identifying this oversight. We have added a description in the figure captions.

-For Figure 6c it should be more clearly described in the main text that the axis for "normalized mean number of divisions" is derived from only the active cell population.

Thank you for this suggestion, we have explicitly written in the figure caption that the mean divisions are computed over active cells exclusively.

Reviewer #1 (Remarks to the Author)

All previously raised points were satisfactorily addressed by the authors. For example suggested additional experiments were performed to prove the existence of unidirectional information processing in the MAPK pathway in T lymphocytes without a negative feedback. Demanded references were included and clarifications were made as requests.

As a small remark, the findings proposed here are only qualitative observations due to the rather coarse-grained modeling approach. Mechanistic links are not discussed any further as the authors consider them beyond scope.

Reviewer #2 (Remarks to the Author)

In this revised submission, the authors have satisfactorily responded to my previous concerns. I find this manuscript to be highly important, interesting, and appropriate for the readership of Nature Communications. Their results reveal how perturbations to early signaling events can impact cell fate decisions over longer timescales, in addition to providing a valuable framework for future studies to quantify mechanisms of drug action in single cells.